# Evaluating Medical Text Summaries Using Automatic Evaluation Metrics and LLM-as-a-Judge Approach: A Pilot Study

**DOI:** 10.3390/diagnostics16010003

**Published:** 2025-12-19

**Authors:** Yuriy Vasilev, Irina Raznitsyna, Anastasia Pamova, Tikhon Burtsev, Tatiana Bobrovskaya, Pavel Kosov, Anton Vladzymyrskyy, Olga Omelyanskaya, Kirill Arzamasov

**Affiliations:** 1Research and Practical Clinical Center for Diagnostics and Telemedicine Technologies of the Moscow Health Care Department, 127051 Moscow, Russia; 2Institute of Artificial Intelligence, MIREA—Russian Technological University, 119454 Moscow, Russia; 3Department of Information Technology and Medical Data Processing, Federal State I.M. Sechenov First Moscow State Medical University of the Ministry of Health of the Russian Federation (Sechenov University), 119991 Moscow, Russia

**Keywords:** large language model, summaries, electronic health records, LLM-as-a-judge

## Abstract

**Background:** Electronic health records (EHRs) remain a vital source of clinical information, yet processing these heterogeneous data is extremely labor-intensive. Summarization of these data using Large Language Models (LLMs) is considered a promising tool to support practicing physicians. Unbiased, automated quality control is crucial for integrating the tools into routine practice, saving time and labor. This pilot study aimed to assess the potential and constraints of self-contained evaluation of summarization quality (without expert involvement) based on automatic evaluation metrics and LLM-as-a-judge. **Methods:** The summaries of text data from 30 EHRs were generated by six open-source low-parameter LLMs. The medical summaries were evaluated using standard automatic metrics (BLEU, ROUGE, METEOR, BERTScore) as well as the LLM-as-a-judge approach using the following criteria: relevance, completeness, redundancy, coherence and structure, grammar and terminology, and hallucinations. Expert evaluation was conducted using the same criteria. **Results:** The results showed that LLMs hold great promise for summarizing medical data. Nevertheless, neither the evaluation metrics nor LLM judges are reliable in detecting factual errors and semantic distortions (hallucinations). In terms of relevance, the Pearson correlation between the summary quality score and the expert opinions was 0.688. **Conclusions:** Completely automating the evaluation of medical summaries remains challenging. Further research should focus on dedicated methods for detecting hallucinations, along with investigating larger or specialized models trained on medical texts. Additionally, the potential integration of retrieval-augmented generation (RAG) within the LLM-as-a-judge architecture deserves attention. Nevertheless, even now, the combination of LLMs and the automatic evaluation metrics can underpin medical decision support systems by performing initial evaluations and highlighting potential shortcomings for expert review.

## 1. Introduction

Using medical history as the primary source of information about patient health is an important step toward accurate diagnosis. Patient interviews do not always deliver comprehensive clinical presentation, and analyzing medical records is labor-intensive and time-consuming.

Large language models (LLMs) are a type of transformer neural networks that have been pre-trained on vast amounts of textual data. LLMs utilize statistical analysis of text and its components as data units or tokens (sequences of words, syllables, and letters). The models predict the most probable continuation for a given sequence of prompt tokens, taking into account syntactic, semantic, and ontological relationships [1]. LLMs are capable of processing large volumes of text and generating human-like summaries and interpretations [2].

LLMs have been drawing increasing attention in medicine as tools for extracting clinically significant information, analyzing and interpreting data, etc. [3,4].

Research confirms the effectiveness of LLMs as clinical decision support tools in specialized fields such as oncology [5], radiology [6], otolaryngology [7], pediatric nephrology [8], and others.

The capabilities of LLMs are also expanding toward data analysis and interpretation, including generating highly accurate recommendations for patient follow-up based on radiological reports and clinical guidelines [9], as well as predicting medication prescriptions through the analysis of clinical information [10]. LLMs facilitate the automation of documentation workflows by summarizing physician consultations based on transcriptions of clinical conversation [11]. They are increasingly integrated into care planning, medical education, and other organizational processes [12].

This work focuses on the application of LLMs for medical text summarization, designed to facilitate the analysis of medical data for specialists in various fields, ensure continuity, and improve the quality of medical care [13].

Today, the integration of AI-powered technologies in medicine is accompanied by understandable mistrust and apprehension. Any LLM poses the risks of illusory conclusions (“hallucinations”), while both input data and evaluation outputs may be biased due to algorithmic limitations and lack of diversity in training datasets. These shortcomings negatively impact the quality of medical summaries, leading to misrepresentation of patient’s condition and ultimately affecting treatment decisions. Therefore, in Russian Federation LLM-based methods (like all AI technologies) are classified in medicine as Class III high-risk technologies [14].

Quality control of LLM performance is critical, as the summaries must comply with the established requirements.

In a number of studies, the quality of LLM summaries is assessed by both domain specialists and established automatic metrics that evaluate the performance of natural language processing: ROUGE [15], BLUE [16], Meteor [17], BERTScore [18], etc. Such studies draw from combinations of various metrics, expert evaluation criteria, principles and assessment scales, and approaches to calculating the final score, where applicable [2,19].

The labor intensity of expert work, the large volumes of analyzable text data, the growing number of LLMs, and the expanding range of clinical challenges require methods and tools to fully or partially automate quality control of model performance.

Autonomous summary assessment using the automatic evaluation metrics alone is impossible, as they only reflect the similarity between two texts: an LLM output and a reference. Typically, a reference summary is created by a human expert in the relevant field. The automatic evaluation metrics do not allow assessment of coherence or readability and are unable to determine if key information is missing; therefore, they lack correlation with expert assessment [2,20].

In searching for an autonomous method for summary evaluation, research has suggested using LLMs as evaluators (LLM-as-a-judge) [21]. LLM-as-a-judge utilizes user-defined criteria (e.g., coherence, consistency, relevance, accuracy) defined in prompts—specific instructions for neural networks.

One advantage of the LLM-as-a-judge approach is that the method does not require reference summaries. Furthermore, some generative models capable of processing large volumes of text data evaluate summaries generated from voluminous texts. Such an approach ensures better detection of substantive errors.

However, only a few studies report high consistency between LLM-as-a-judge and expert outputs that overall met the researchers’ needs [22,23]. Inherent LLM biases also impact the performance of LLM-as-a-judge. Therefore, biased assessments, preference for redundancy at the expense of accuracy, overestimation of the decision quality, and similar issues [24] prevent LLM judges from being considered reliable, autonomous evaluation tools, especially in the highly sensitive medical field. Currently, LLM-as-a-judge has not yet achieved expert-level evaluation of clinical summaries, despite existing bias-mitigating methods and other reliability-enhancing strategies [25,26].

The aforementioned works primarily employ proprietary models (e.g., Claude, Gemini, ChatGPT). However, legislative constraints on personal data protection limit their applicability in healthcare, as no EHR anonymization tool can guarantee absolute privacy preservation.

Recent research further justifies a focus on smaller, specialized models. For instance, MedHELM demonstrates that for certain medical tasks, particularly summarization, such models can outperform large general-purpose ones [27]. Similarly, A. Ahmed et al. propose enhancing low-parameter models for medical contexts through advanced prompting techniques, improving their accuracy and relevance [28]. However, independent fine-tuning of models requires substantial resources, primarily due to expert data annotation. Models trained on narrow datasets also tend to exhibit poor scalability.

Motivated by the need to reduce these costs, there is considerable scientific and practical interest in evaluating the potential of publicly available, low-parameter general-purpose models to solve domain-specific tasks on limited hardware.

The primary objective of this work is to determine whether existing approaches to the automated evaluation of generated texts can be adapted to the task of medical text summarization under constraints of limited computational resources and budget. This study aims to identify the main challenges and propose further strategies for employing small, open-access LLM-as-a-judge.

This paper presents the results of a pilot study assessing the quality of medical text summarization using a set of automatic evaluation metrics and open-source LLM-as-a-judge. In addition, an analysis of the agreement between the integrated summarization quality assessment results and expert evaluations was conducted.

## 2. Materials and Methods

This study assessed the summaries of 30 text documents containing electronic health record (EHR) data. The summaries were generated to extract data relevant to radiological reports. We used six state-of-the-art generative models, varying in size (number of parameters), quantization format, and reasoning mode. The reference summaries were created by clinicians and validated by a radiologist. The summaries generated by the LLMs were assessed using the automatic evaluation metrics, two LLM-as-a-judges, and experts.

Thus, the pilot study investigated 180 summaries using each of the criteria below. This sample size allowed us to draw preliminary conclusions regarding the validity of the automatic evaluation metrics and the scores assigned by the LLM judges. This made it possible to evaluate their contribution to the integrated score and the ability to reproduce expert decisions. In addition, it allowed identification of promising areas for using a comprehensive criterion representing the quality of medical text summarization.

All experiments were run on a dual RTX 3090 Ti setup (24 GB VRAM per GPU) (NVIDIA corp., Santa Clara, CA, USA), which offered sufficient performance for inference of the LLMs.

The study was conducted as part of the research and development project “A promising automated workplace of a radiologist based on generative artificial intelligence.” The study protocol was approved by the IEC of the Moscow Regional Branch of the Russian Society of Roentgenographers and Radiologists (MRB RSRR) (Protocol No. 6 dated 19 June 2025).

### 2.1. Dataset

The dataset consisted of 30 text files (from 1450 to 4216 words) generated from EHR data. The inclusion criteria for EHR were the availability of the following information:•Two radiology reports for the same anatomical region, acquired with the same modality;•Examination report created by the specialist who referred the patient for the last of these imaging studies;•A report for an additional imaging study acquired using a different modality or on a different anatomical area;•Two values of a lab test (initial and follow-up);•An additional lab test value different from the above;•An examination report by a physician of any specialty not related to imaging studies;•A hospital discharge summary.

Only EHRs belonging to patients over 18 years of age were included in the study.

### 2.2. LLMs for Summarizers and LLMs-as-a-Judge

Models hosted on the Hugging Face platform [29] were used to summarize the EHR data. LLM inclusion criteria:•Released no earlier than Q1 2024;•Available under the Apache 2.0 open-source license;•Architecture type: transformer;•Capable of handling large volumes of text data.•Deployment environment: Ollama, 2 RTX 3090 Ti GPUs

The resulting six LLMs are presented in Table 1.

The Q4 label in the LLM name denotes the use of the level 4 quantization method q4_K_M, which reduces the size of neural network weights by reducing the bit depth of floating-point numbers. This method secures balance between the speed and accuracy of LLMs.

The nt label denotes the use of the “no_think” mode. This mode disables the LLM’s analytical capabilities, eliminating the abstract reasoning and intermediate checks. Such models run faster but produce less detailed and in-depth answers.

All models were run at zero temperature, which minimizes generation stochasticity and ensures maximum correspondence between the generated summaries and the EHR data.

Two models from different families were selected to act as LLM-as-a-judge: Qwen3-14b and Mistral-small-24b. The first model was chosen to test the hypothesis about the estimate bias caused by testing the model on its own data. The evaluation utilized the same criteria as the expert review. The query for the LLM judges was an adaptation of the expert query presented in Appendix A.

No specialized fine-tuning of the models was performed. It reflects a practical scenario where end-users deploy off-the-shelf models without adaptation, which improves the reproducibility and generalizability of our methodology.

### 2.3. Reference Summary and Expert Review

Four expert physicians (three clinicians and one radiologist) with over three years of experience provided reference summaries for 30 texts. The task was designed to extract the data that radiologists could use in abdominal computed tomography (CT) interpretation, namely complaints underlying the referral, medical and life history (comorbidities, bad habits, family history, surgeries), as well as data from lab tests and imaging studies.

The summaries were to contain nothing but information clinically relevant to abdominal CT [30].

Eighteen experts evaluated the summarization performance. Three experts and a radiologist assessed each generated summary according to the following criteria: relevance, recall, redundancy, coherence and structure, grammar and terminology, hallucinations [31].

The experts used a binary scale (present/absent) to assess the “Hallucinations” criterion. The resulting binary values were converted into quantitative scores: 1 point was assigned to summaries containing a hallucination, and 5 points were assigned if no hallucination was present. The remaining criteria were assessed on a five-point Likert scale, where 1 corresponded to the worst result and 5 to the best.

A breakdown of the criteria is provided in Appendix A.

The final score was determined by expert consensus.

### 2.4. Automatic Evaluation Metrics

The automatic evaluation metrics that measure summary quality are based on similarity to the reference summary. In this paper, the following metrics were analyzed:ROUGE (Recall-Oriented Understudy for Gisting Evaluation). This class of metrics evaluate the recall and accuracy of summarization on a scale from 0 to 1 by analyzing the match of n-grams and their sequences with a reference summary.

ROUGE-1 evaluates the match of unigrams (individual words). This is the simplest measure, as it does not consider word order.

ROUGE-2 evaluates the match of bigrams (pairs of consecutive words). This metric takes into account word order but is limited to short sequences.

ROUGE-L, unlike the previous metrics, focuses on the longest common subsequences in the generated and reference summaries. ROUGE-L takes into account the consistency of longer text fragments, allowing assessment of the structural similarity between sentences.

2.BLEU (Bilingual Evaluation Understudy) uses a weighted combination of n-gram matches to estimate similarity and introduces a brevity penalty. This approach enables the comparison of generated texts with multiple reference summaries to ensure a more accurate assessment.3.METEOR (Metric for Evaluation of Translation with Explicit Ordering) is a modification of ROUGE that accounts for the variability of the words’ morphological features through stemming and lemmatization. The algorithm penalizes unrelated fragments, permutations, and duplication.4.BERTScore (Bidirectional Encoder Representations from Transformers) evaluates the similarity between generated and reference summaries by calculating the semantic distance between individual word vectors (tokens) using a pre-trained BERT model. Unlike previous evaluation metrics, BERTScore takes into account the semantic and syntactic similarity between the generated and the reference summaries but ignores key quality criteria such as coherence and factual accuracy. The method is computationally intensive and relies on the quality of pre-trained BERT models.

### 2.5. Statistical Analysis

Statistical analysis was performed using Python 3.12 software with the SciPy and pandas libraries (Python Software Foundation, Wilmington, DE, USA) and JASP (v. 0.19.3.0) (JASP Team, Amsterdam, The Netherlands). All evaluations considered statistical significance α = 0.05.

We analyzed the mean (M), the standard deviation (SD), the median (Me), and the interquartile range [Q1; Q3]. The normal distribution of quantitative variables was tested using the Shapiro–Wilk test. The significance of differences was determined using the Wilcoxon signed-rank test.

The strength and direction of the relationship between two quantitative variables were assessed using the Spearman correlation coefficient (r_s_). We used multiple linear regression analysis to test the hypothesis regarding the predictability of expert scores based on the evaluation metrics and the LLM-as-a-judge outputs.

## 3. Results

The average summary generation time was 25 s. In contrast, the automated evaluation by Qwen3-14b took, on average, 200 s, and that by Mistral-small-24b took 85 s.

Table 2 presents the expert scores for 30 summaries for each LLM.

The expert scores illustrated the LLM capabilities in summarizing EHR data. The Llama family models with 70 billion and 8 billion parameters demonstrated consistently high summarization performance across all evaluation criteria. Interestingly, the Gemma model (27 billion) demonstrated lower overall performance on average compared to the Llama model (8 billion), with the exception of the “Grammar and Terminology” criterion. The Qwen model (32 billion) demonstrated a tendency to generate complete yet redundant summaries compared to the more compact Llama model (8 billion). Compared to Qwen (32 billion), the Gemma model (12 billion) showed significantly lower relevance and completeness while outperforming in grammar and terminology, with similarly low hallucinations rate.

Thus, summarization quality assessment requires a multi-criteria approach, and the model performance can vary depending on the task and, accordingly, the priority of a particular criterion.

For practical purposes, agreement between the absolute values of metrics from LLMs and experts can be neglected. The validity criterion for LLM-as-a-judge is the adequate summary ranking. In other words, an LLM judge should assign lower scores to lower-quality summaries.

Therefore, to select an LLM judge, we analyzed the correlation of metrics obtained by experts and the LLM judges (Figure 1) using Spearman’s rank correlation coefficient.

We also tested the hypothesis that the self-assessing model generates overestimated metric values (Table 3). For this purpose, we compared metrics that reflect the Qwen3-32B_Q4-nt performance using two LLM judges. Average metric values were used for comparison, as median values are uninformative. The significance of differences across all scores given by the LLM judge (except for hallucinations) was assessed with the Wilcoxon *t*-test.

The remaining summaries revealed Qwen3-14b significantly overestimated the relevance, completeness, and redundancy scores, compared to Mistral-small-24b. However, both LLMs identified only two of the nine hallucinations detected by the experts.

Therefore, the Qwen3-14b LLM metrics were excluded from further analysis due to lower correlation with the expert assessment compared to Mistral-small-24b (Figure 1), in addition to the bias manifested as score overestimation for the models of its class (Table 3).

Another observation was the inability of the LLM judges to accurately estimate the summary quality in terms of “Redundancy”, “Coherence, Structure”, “Hallucinations”, and “Grammar, Terminology”. This confirms the shortcomings noted in the literature and demonstrates the infeasibility of using LLM judges as standalone evaluators without external support such as that provided by automatic evaluation metrics.

### 3.1. Performance of the Automatic Evaluation Metrics

Similarly, we analyzed the automatic evaluation metrics’ performance and their correlation with expert assessments. Since the evaluation metrics are not directly comparable with expert criteria, the pairwise correlation of all metrics was analyzed. The results are presented in Table 4.

The expectedly low correlation between the automatic evaluation metrics and the expert scores provides evidence of their infeasibility as reliable independent evaluators. These results were not unexpected: evaluation metrics are only capable of assessing similarity to a reference summary, one of many that meet the criteria. Meanwhile, experts evaluate the quality of summaries without needing the reference. Moreover, the BERTScore metric demonstrated a stronger correlation with the expert scores, as it takes into account possible synonymous substitutions and syntactic paraphrases, demonstrating robustness to lexical diversity.

To analyze the relationship between the automatic evaluation metrics and the LLM judges, a correlation analysis was conducted. The results are presented in Figure 2.

The results demonstrated low correlation between the metrics, which indicates a weak relationship. This led us to conclude that each group of metrics evaluates the summary quality from different perspectives. Therefore, a standalone assessment requires using both metric groups, which do not duplicate but rather complement each other, providing a comprehensive summary quality score.

### 3.2. Standalone Integrated Assessment of Summarization Performance

To develop an integrated score, we relied on multiple linear regressions using the elimination method to remove the least significant predictors from a model that initially included all possible predictors. The Mistral-small-24b model was selected as the LLM judge.

Table 5 presents the three most significant predictors for each criterion score. Multiple correlation coefficients (R) were calculated. The correlation coefficients for BERTScore, the corresponding LLM judge metric, and the multiple correlation coefficient of the model that incorporated a full stack of predictors are provided for comparison. The R^2^ coefficient of determination and the root mean square error (RMSE) for the full model are shown.

The results of the correlation analysis show the expected increase in metrics for a model that incorporated a full stack of predictors. It is noteworthy that among the most significant predictors for expert scores, the BERTScore and Relevance criteria were the most frequent.

According to the correlation analysis, the automatic evaluation metrics and the LLM judges’ scores demonstrated the highest correlation with the expert scores in terms of Relevance and Recall. It is noteworthy that Relevance can be considered a general score or a sum of scores reflecting the summarization performance, which cannot be high given low scores for any other criterion. This allows Relevance to be considered an integrated score of the summary quality.

This assertion is supported by the fact that Relevance measured by the LLM judge has the greatest predictive power for expert decisions on four of the six criteria.

## 4. Discussion

Relatively high expert scores were observed for “Grammar, Terminology” and “Coherence, Structure” across most of the tested models. All selected LLMs demonstrated the ability to generate summaries that met the high standards set by the established parameters. The low correlation between the expert scores and LLM-as-a-judge may be explained by both minor differences in the quality of summaries for these criteria and limitations of expert review. Evaluations derived from expert opinion are inherently subjective, being grounded in individual professional experience. Consequently, within the scope of this study, perfect agreement between expert assessments and the automated approach across all metrics—”Completeness”, “Relevance”, “Redundancy”, and “Coherence, Structure”—cannot be expected. However, the identification of factual inaccuracies remains critical.

The LLM judges effectively identify significant shortcomings in the summaries; however, they frequently conflate or misapply the evaluation criteria. Illustrative comments explaining a score reduction on a specific criterion are provided in Table 6.

Interestingly, BERTScore demonstrated a significantly higher correlation with expert opinions compared to LLM-as-a-judge.

The key reason may lie in the fundamental difference in the approaches to the scoring. An LLM judge, even when asked to abide by strict criteria, remains a “generalist” whose decisions are influenced by the vast array of data on which it was trained. While models simulate reasoning, they suffer from high entropy and are biased toward superficial text attributes. Furthermore, our experiment used a quantized version of Mistral Small 24B, which in itself reduces calculation accuracy and decreases sensitivity to subtle semantic differences.

The large volume and complexity of the processed texts significantly impacted the summarization performance. As is well known, modern LLMs suffer from “contextual forgetting”: as the text length increases, model performance worsens and relevant information gets lost [32,33,34].

At the same time, the BERTScore architecture is designed to process text fragments using semantic frame embeddings, maintaining accuracy even with large context lengths. Thus, BERTScore is a “special tool” optimized for one specific problem: determining the extent to which the semantics of a single text corresponds to a larger semantic space optimized for the medical domain.

When the reference text is produced by an expert, BERTScore measures the degree of conformity to expert expectations through domain-specific embeddings, whereas Mistral Small 24B, being an open quantized model with limited domain adaptation, cannot compete with the target metrics in settings requiring high factual accuracy.

Thus, the generality and complexity of LLMs do not guarantee superiority in narrow scoring tasks. Furthermore, for tasks requiring high factual and semantic accuracy, simpler and more specialized tools may prove significantly more effective.

It should also be noted that neither the proposed automatic evaluation metrics nor LLM judges are capable of reliably detecting hallucinations, which occur in LLMs even at zero temperature for summary generation. For medical tasks, this aspect is critical. Errors in the summaries of medical history, laboratory data, and drug therapy can mislead physicians, distort the clinical presentation, and ultimately harm patients.

The analysis revealed several factual inaccuracies in LLM-generated summaries. For example, instances of unacceptable fabrication were observed. In one case, a summary produced by the Gemma-2-27b-it model included data on patient alcohol abuse—information absent from the source text. In an effort to construct a coherent medical narrative, the model apparently substituted the missing detail with a statistically probable inference derived from patterns in its training data. This hallucination was correctly identified by LLM-as-a-judge, which flagged it as a specific factual error.

Errors stemming from incorrect data interpretation were also noted. While the original EHR contained a referral for an abdominal ultrasound, the corresponding summary stated “Abdominal ultrasound: not performed”—a claim potentially misrepresenting the actual course of events. This discrepancy was also successfully detected.

Finally, LLMs demonstrated notable weaknesses in numerical analysis. Consequently, the LLM-as-a-judge failed to detect a significant error where the reported body temperature was inaccurately stated as 39.0 °C instead of the correct 36.6 °C. Such a numerical substitution is clinically important.

Modern literature addresses the problem of automated approaches to assessing the credibility of generated texts. Methods for automated detection of hallucinations based on pre-trained models have been proposed [35,36,37]. While their potential efficiency is high, these methods are limited in their scope of application and require labor-intensive collection and labeling of data specific to each task. Promising directions are general methods that are not tied to particular domains and do not require training on labeled data. For example, Farquhar S. et al. in the article [37] propose a method based on semantic entropy, which provides a partial solution for identifying errors associated with confabulations. The method is not tied to a specific task and could potentially complement the proposed integrated score for the quality of medical text summaries.

Verga P. and colleagues propose an approach that employs a diverse panel of models. This method demonstrates high agreement with expert evaluations in assessing answer quality, achieving a Pearson correlation coefficient of 0.917, compared to 0.817 for GPT-4 [38].

## 5. Limitations and Future Research

This study employed only low-parameter, local, open-access general-purpose models without fine-tuning, as they present the most straightforward and transparent option for practical deployment. Such models can be easily integrated into existing medical information infrastructures: implementation requires installing a local inference server using Ollama, after which summarization may be obtained via a simple HTTP request. The latest Ollama version includes a dedicated client, which simplifies operations and eliminates the need for tools like Jupyter notebooks. Future work could incorporate domain-specific models trained on medical corpora without significant technical refinements.

The research was conducted on a limited dataset using a constrained set of LLMs for both summarization and evaluation. Optimization of model and prompt, followed by validation on a larger dataset, could help identify the most effective approaches for the autonomous assessment of medical summaries.

It should be noted that any changes in expert evaluation methodology—such as criteria, rating scales, or volume and structure of the source text—will influence the resulting metrics. Therefore, full agreement of automated evaluation scores with specific expert benchmarks (or with metrics from other studies) may not be achievable. A more promising objective is to improve the detection of factual errors through novel methodological enhancements.

Another direction for future research involves integrating retrieval-augmented generation (RAG) into the LLM judge architecture. RAG can enhance evaluation objectivity and grounding by retrieving relevant context from an external knowledge base.

Further analysis of the influence of hyperparameters, temperature, and model size on the ability to generate and detect hallucinations is also needed.

## 6. Conclusions

Despite significant progress in the field of LLMs, full automation of the assessment of the medical text summarization performance remains an open question, primarily because neither automatic evaluation metrics nor LLM judges are capable of reliably identifying factual errors and semantic distortions (hallucinations).

At the same time, high expert scores indicate the significant potential of LLMs for medical text summarization tasks. Even open-source architectures with small models have demonstrated the ability to adequately systematize and generalize complex and heterogeneous medical data, indicating great potential for further development.

It is worth noting that the development of an autonomous quality assessment tool does not require full correlation with expert opinion, which appears methodologically challenging due to subjectivity of the latter. Implementation of tailored hallucination detection methods and the use of LLMs with a larger number of parameters or more highly specialized LLMs trained primarily on medical data will improve the reliability, validity, and accuracy.

Nevertheless, even at the current stage, the combination of LLMs and the automatic evaluation metrics can provide the basis for medical decision support systems to improve experts’ performance by drawing attention to identified deficiencies in medical summaries

## Figures and Tables

**Figure 1 diagnostics-16-00003-f001:**
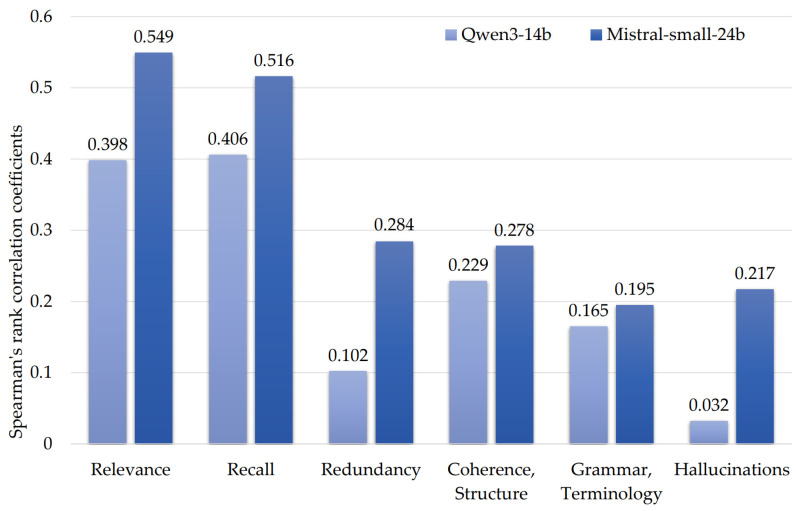
Spearman’s rank correlation coefficients for LLM judges’ metrics and expert scores.

**Figure 2 diagnostics-16-00003-f002:**
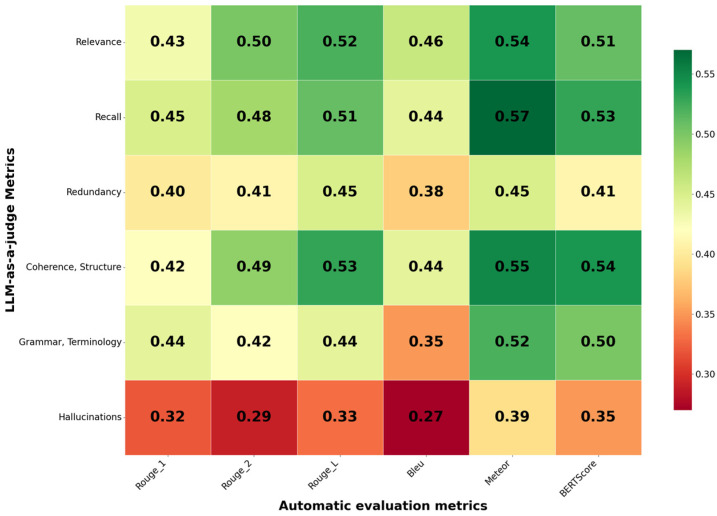
Spearman’s correlation coefficients for the automatic evaluation metrics and Mistral-small-24b.

**Table 1 diagnostics-16-00003-t001:** The six LLMs utilized for summarization in this study.

LLM	Model Size (Parameters)	Developer
Gemma-2-27b-it	27 bln	Google LLC, Mountain View, CA, USA
Llama-3.3-70B-Instruct_Q4	70 bln	Meta Platforms Inc., Menlo Park, CA, USA
Llama3.1-8b	8 bln	Meta Platforms Inc., Menlo Park, CA, USA
Llama-3.2-3B-Instruct	3 bln	Meta Platforms Inc., Menlo Park, CA, USA
Gemma-3-12b-it	12 bln	Google LLC, Mountain View, CA, USA
Qwen3-32B_Q4-nt	32 bln	Qwen Ltd., Beijing, China

**Table 2 diagnostics-16-00003-t002:** Expert scores of the summary quality for each LLM. M ± SD for each indicator are shown.

Model/Criteria	Gemma-2-27b-it	Llama-3.3-70B-Instruct_Q4	Llama3.1-8b	Llama-3.2-3B-Instruct	Gemma-3-12b-it	Qwen3-32B_Q4-nt
Relevance	2.6 ± 1.3	3.8 ± 1.0	3.3 ± 0.9	2.4 ± 1.0	3.6 ± 1.2	3.8 ± 1.3
Recall	2.6 ± 1.3	3.7 ± 1.0	3.1 ± 1.1	2.3 ± 1.1	3.9 ± 1.1	3.9 ± 1.2
Redundancy	3.2 ± 1.4	4.2 ± 0.8	3.5 ± 1.1	3.2 ± 1.2	4.0 ± 1.0	3.2 ± 1.3
Coherence, Structure	3.6 ± 1.2	4.4 ± 1.0	3.6 ± 1.2	3.1 ± 1.1	4.2 ± 1.0	4.2 ± 1.2
Grammar, Terminology	4.5 ± 0.6	4.7 ± 0.7	4.4 ± 0.8	3.6 ± 1.2	4.7 ± 0.5	4.3 ± 1.1
Hallucinations	3.8 ± 1.9	4.6 ± 1.2	3.6 ± 1.9	3.4 ± 2.0	3.9 ± 1.8	3.8 ± 1.9

**Table 3 diagnostics-16-00003-t003:** Evaluation of Qwen3-32B_Q4-nt summaries of 30 texts. M ± SD and *p*-values (Wilcoxon test for group comparisons) are shown.

Criteria/ LLM-as-a-Judge	Relevance	Recall	Redundancy	Coherence, Structure	Grammar, Terminology
Qwen3-14b M ± SD Me [Q1; Q3]	5.0 ± 0.0 5 [5; 5]	4.8 ± 0.6 5 [5; 5]	5.0 ± 0.0 5 [5; 5]	5.0 ± 0.0 5 [5; 5]	5.0 ± 0.0 5 [5; 5]
Mistral-small-24b M ± SD Me [Q1; Q3]	4.4 ± 0.8 5 [5; 5]	4.2 ± 0.7 4 [4; 5]	3.6 ± 0.8 4 [3; 4]	4.8 ± 0.4 5 [5; 5]	4.8 ± 0.4 5 [5; 5]
*p*-value	<0.001	0.001	<0.001	0.014	0.025

**Table 4 diagnostics-16-00003-t004:** Spearman’s correlation coefficients for the automatic evaluation metrics and expert scores.

	ROUGE-1	ROUGE-2	ROUGE-L	BLEU	METEOR	BERTScore
Relevance	0.44	0.45	0.46	0.45	0.48	0.57
Recall	0.40	0.41	0.40	0.37	0.49	0.53
Redundancy	0.35	0.29	0.31	0.27	0.21	0.33
Coherence, Structure	0.30	0.35	0.32	0.31	0.27	0.42
Grammar, Terminology	0.28	0.30	0.31	0.28	0.21	0.29
Hallucinations	0.17	0.10	0.19	0.11	0.14	0.23

**Table 5 diagnostics-16-00003-t005:** Multiple correlation coefficients R of various models with expert score metrics. The R^2^ coefficient of determination and the root mean square error (RMSE) for the model incorporated a full stack of predictors are shown.

Expert Score Metric	Metrics with the Greatest Impact	R (Three Metrics)	R (BERTScore)	R (LLM-as-a-Judge)	R (Full Model)	R^2^ (Full Model)	RMSE (Full Model)
Relevance	BERTScore	0.673	0.610	0.585	0.688	0.474	0.944
	Relevance (LLM-as-a-judge)						
	Hallucinations (LLM-as-a-judge)						
Recall	BERTScore	0.502	0.377	0.281	0.528	0.278	1.070
	Relevance (LLM-as-a-judge)						
	Meteor						
Redundancy	Rouge_1	0.502	0.377	0.281	0.528	0.278	1.070
	Relevance (LLM-as-a-judge)						
	Meteor						
Coherence, Structure	Rouge_2	0.437	0.406	0.224	0.471	0.222	1.072
	Meteor						
	BERTScore						
Grammar, Terminology	Rouge_L	0.420	0.377	0.247	0.463	0.214	0.826
	Meteor						
	BERTScore						
Hallucinations	Rouge_2	0.306	0.226	0.217	0.355	0.126	1.750
	Rouge_L						
	BERTScore						

**Table 6 diagnostics-16-00003-t006:** Explanation by the LLM-as-a-judge of a score reduction relative to a specific criterion, and the correct criterion associated with the identified shortcoming.

LLM-as-a-Judge Comment	Criterion for Which the LLM-as-a-Judge Lowered the Score	Correct Criterion
The summary contains a lot of information, but it is not always logically structured or connected to the main objective—the description of an abdominal CT scan. The text includes data not relevant to this task, such as a detailed description of the cardiovascular system’s condition, treatment, and recommendations related to paroxysmal atrial fibrillation. This creates unnecessary cognitive load and may hinder the radiologist’s ability to extract the necessary information.	Coherence, Structure	Relevance, Redundancy
The statement “Male, age not specified” does not correspond to the source text, which states the age as 77 years. Additionally, there are repetitions in the “Recommendations” section, making the text less structured.	Grammar, Terminology	Hallucinations, Coherence, Structure
It is claimed, “the patient is in serious condition, “which is not substantiated in the source text. The source text describes only symptoms (pain, weakness) without an assessment of the general condition as “serious.”	Grammar, Terminology	Hallucinations, Relevance
The summary includes the statement “harmful habits: smoking, alcohol consumption (unspecified).” This information is absent from the source text and represents an unwarranted addition. Moreover, assertions regarding the “patient’s serious condition” and the “necessity of an urgent CT scan” are interpretative in nature and are not directly derived from the factual data presented in the original text.	Redundancy	Hallucinations, Relevance

## Data Availability

The raw data supporting the conclusions of this article will be made available by the authors on request.

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
