# Peer review of "Evaluating Medical Text Summaries Using Automatic Evaluation Metrics and LLM-as-a-Judge Approach: A Pilot Study"

_diagnostics, 2025, doi:10.3390/diagnostics16010003_

Round 1
Reviewer 1 Report
Comments and Suggestions for Authors
The introduction is sufficient, but I recommend adding a sentence or two to explicitly state the primary research question or hypothesis of this pilot study for enhanced clarity.
I suggest preparing a table that outlines key factors differentiating existing works from the proposed research. This will provide a clearer context for your contributions.
Consider adding more practical guidance on implementation. Discuss aspects such as computational costs, estimated running time, and ease of integration into existing systems. Additionally, address potential biases in training data and their impact on accuracy.
Although the quantitative results are robust, the qualitative analysis receives limited discussion. I recommend incorporating detailed examples.
Please add limitations and future work.
I encourage you to consider adding the following references when discussing the potential applications of LLMs in the medical field:
- MED-Prompt: A novel prompt engineering framework for medicine prediction on free-text clinical notes.
- Prompt-eng: Healthcare prompt engineering: Revolutionizing healthcare applications with precision prompts.
The abbreviation list appears incomplete, with only four entries added. Please revisit and include all necessary entries to enhance comprehension.
Author Response
Dear Reviewer,
We thank you for your attention to our article and for your valuable comments. We have carefully considered each of them and have improved the manuscript accordingly.
Comments 1: The introduction is sufficient, but I recommend adding a sentence or two to explicitly state the primary research question or hypothesis of this pilot study for enhanced clarity.
Response 1: This information has been added at the end of the Introduction section (lines 120-124).
Comments 2: I suggest preparing a table that outlines key factors differentiating existing works from the proposed research. This will provide a clearer context for your contributions.
Response 2: We have concisely formulated the critical distinctions between the cited studies and our approach (the use of open-source low-parameter models without fine-tuning), rephrasing our core idea in a more clear manner (lines 106-119).
Comments 3: Consider adding more practical guidance on implementation. Discuss aspects such as computational costs, estimated running time, and ease of integration into existing systems. Additionally, address potential biases in training data and their impact on accuracy.
Response 3: We agree with this point. A dedicated section titled "Limitations and Future Research" has been added (line 426), which includes a description of practical considerations for model deployment. We have also enriched the article with practice-oriented information (lines 252-254, Table 6, 397-410).
Comments 4: Although the quantitative results are robust, the qualitative analysis receives limited discussion. I recommend incorporating detailed examples.
Response 4: Several illustrative examples and clarifications have been added to the "Discussion" section (lines 252-254, Table 6).
Comments 5: Please add limitations and future work.
Response 5: The section “Limitations and future work” has been added (line 426).
Comments 6: I encourage you to consider adding the following references when discussing the potential applications of LLMs in the medical field:
• MED-Prompt: A novel prompt engineering framework for medicine prediction on free-text clinical notes.
• Prompt-eng: Healthcare prompt engineering: Revolutionizing healthcare applications with precision prompts
Response 6: We thank you for providing this list of references. The mentioned articles have been incorporated into the literature review (Ref. 10, 28).
Comments 7: The abbreviation list appears incomplete, with only four entries added. Please revisit and include all necessary entries to enhance comprehension.
Response 7: The missing abbreviations have been added to the relevant section.
Reviewer 2 Report
Comments and Suggestions for Authors
Electronic health records (EHRs) remain a vital source of clinical information, yet processing these heterogeneous data is extremely labor-intensive. Summarization of these data using Large Language Models (LLMs) is considered a promising tool to support practising physicians. Unbiased, automated quality control is crucial for integrating the tools into routine practice, saving time and labor.
AUTHORS propose a pilot study aimed to assess the potential and constraints of self-contained evaluation of summarization quality (without expert involvement) based on automatic evaluation metrics and LLM-as-a-judge.
THEIR methodology proposed the following approach:
- The summaries of text data from 30 EHRs were generated by six open-source LLMs.
- The medical summaries were evaluated using standard automatic metrics (BLEU, ROUGE, METEOR, BERTScore) as well as the LLM-as-a-judge approach using the following criteria: relevance, completeness, redundancy, coherence and structure, grammar and terminology, and hallucinations.
- Expert evaluation was conducted using the same criteria.
The results showed that LLMs hold great promise for summarizing medical data. Nevertheless, neither the evaluation metrics nor LLM judges are reliable in detecting factual errors and semantic distortions (hallucinations). In terms of relevance, the Pearson correlation between the summary quality score and the expert opinions was 0.688.
AUTHORS concluded that:
- Completely automating the evaluation of medical summaries remains challenging. Further research should focus on dedicated methods for detecting hallucinations, along with investigating larger or specialized models trained on medical texts.
- Nevertheless, even now, the combination of LLMs and the automatic evaluation metrics can underpin medical decision support systems by performing initial evaluations and highlighting potential shortcomings for expert review.
The study is truly interesting and timely, and makes a truly important contribution to the scientific literature.
I have the following comments for the authors:
- In the introduction, please expand on the section on the LLMs and explain to the reader what they are based on. I think this is important background information.
- Some sections of the introduction should be further developed, explaining the contribution of individual studies. See, for example, the passage "Today, LLMs are incorporated into routine radiology, particularly for automated radiological reporting and medical data extraction [4–6]."
- The methods consist of seven coherent and scientifically valuable sections. However, they could be improved by avoiding excessive use of lists, streamlining the text (for example, lines 170–184), and inserting tables where possible (for example, lines 146–154).
- Some very short sections of the methods can be merged
- The results are interesting; remember to include data labels in the histograms.
- The discussion is actually missing: insert a discussion with comparison with other works and with the limitations of the study
Author Response
Dear Reviewer,
We thank you for your attention to our study and for your valuable comments. We have carefully considered each of them and have improved the manuscript accordingly.
Comments 1: In the introduction, please expand on the section on the LLMs and explain to the reader what they are based on. I think this is important background information.
Response 1: The corresponding section has been revised, and a relevant literature reference has been added (lines 44-50).
Comments 2: Some sections of the introduction should be further developed, explaining the contribution of individual studies. See, for example, the passage "Today, LLMs are incorporated into routine radiology, particularly for automated radiological reporting and medical data extraction [4–6]."
Response 2: The literature review has been expanded, with references now accompanied by specific explanations of the contributions of those works (lines 53-62).
Comments 3: The methods consist of seven coherent and scientifically valuable sections. However, they could be improved by avoiding excessive use of lists, streamlining the text (for example, lines 170–184), and inserting tables where possible (for example, lines 146–154).
Response 3: The information has been restructured in accordance with this comment for better clarity (lines 173,199-202).
Comments 4: Some very short sections of the methods can be merged.
Response 4: Several sections have been consolidated into one (lines 164,191).
Comments 5: The results are interesting; remember to include data labels in the histograms.
Response 5: We have reviewed all graphical material for missing data labels and have not identified any deficiencies. If this comment remains unaddressed, we kindly request clarification regarding the specific figure number and what information you believe should be added.
Comments 6: The discussion is actually missing: insert a discussion with comparison with other works and with the limitations of the study.
Response 6: We have added the corresponding section, including information on how various methodological aspects of the study influenced the obtained results (422-425, 439-444).
I sincerely appreciate you taking the time to review this and for your insightful comments.
Round 2
Reviewer 1 Report
Comments and Suggestions for Authors
Dear Authors,
Thank you for addressing all my comments. The current version is acceptable for publication, but I have identified a few minor issues that requires corrections:
-
Please reference in numeric ascending order, such as [2, 19] instead of [19, 2]. Kindly revisit all citations to ensure consistency.
-
I noticed a formatting issue with extra spaces in the text, particularly between lines 112-115.
-
The sentence at line 190 requires attention: “No specialized fine-tuning of the models was performed.” It seems a bit too brief—consider expanding on this point.
-
In the text, please consider using complete terms, such as "seconds," instead of abbreviations. For example, revise "The average summary generation time was 25 sec" to "The average summary generation time was 25 seconds."
Author Response
Dear Reviewer,
Thank you for your quick and constructive review.
We are grateful for your helpful observations, which have significantly improved our work. We have incorporated all your suggested changes and have also performed a round of cosmetic edits to polish the manuscript.
-
Please reference in numeric ascending order, such as [2, 19] instead of [19, 2]. Kindly revisit all citations to ensure consistency.
Response: We have fixed this error and revisited all citations (line 80) -
I noticed a formatting issue with extra spaces in the text, particularly between lines 112-115.
Response: We examined the document with formatting marks turned on and fixed several formatting mistakes (lines 52, 114, 198). -
The sentence at line 190 requires attention: “No specialized fine-tuning of the models was performed.” It seems a bit too brief—consider expanding on this point.
Response: We have added a clarifying sentence (lines 190-192) . -
In the text, please consider using complete terms, such as "seconds," instead of abbreviations. For example, revise "The average summary generation time was 25 sec" to "The average summary generation time was 25 seconds."
Response: Comment noted (lines 254-256)
Once again, thank you for your time and constructive review